Metagenomic assembly is the main bottleneck in the identification of mobile genetic elements

Kerkvliet Jesse J. 1
Bossers Alex 2 3
Kers Jannigje G. 2
Meneses Rodrigo 1
Willems Rob 1
Schürch Anita C. a.c.schurch@umcutrecht.nl 1
1 Department of Medical Microbiology, UMC Utrecht , Utrecht , The Netherlands
2 Utrecht University, Institute for Risk Assessment Sciences , Utrecht , The Netherlands
3 Wageningen University, Wageningen Bioveterinary Research , Lelystad , The Netherlands
Zhang Xin
Electronic publication date: 2024 Jan 4
Publication date: 2024
Volume: 12
Electronic Location ID: e16695
Received 2023 Sep 4; Accepted 2023 Nov 28
Copyright: ©2024 Kerkvliet et al.
Copyright year: 2024
Copyright holder: Kerkvliet et al.
License: This is an open access article distributed under the terms of the Creative Commons Attribution License, which permits unrestricted use, distribution, reproduction and adaptation in any medium and for any purpose provided that it is properly attributed. For attribution, the original author(s), title, publication source (PeerJ) and either DOI or URL of the article must be cited.
License URL: https://creativecommons.org/licenses/by/4.0/

Keywords: Metagenomics, Drug resistance, Bacterial, Bacteria, Plasmid, Computational biology

Funding: The Netherlands Center for One Health (NCOH) and the collaboration project ‘DiSSeMINATE’ (LSHM19183) The PPP Allowance made available by Health Holland, Top Sector Life Sciences & Health This research was funded by The Netherlands Center for One Health (NCOH) and the collaboration project ‘DiSSeMINATE’ (LSHM19183) is co-funded by the PPP Allowance made available by Health Holland, Top Sector Life Sciences & Health, to stimulate public-private partnerships. The funders had no role in study design, data collection and analysis, decision to publish, or preparation of the manuscript.

==============================
Antimicrobial resistance genes (ARG) are commonly found on acquired mobile genetic elements (MGEs) such as plasmids or transposons. Understanding the spread of resistance genes associated with mobile elements (mARGs) across different hosts and environments requires linking ARGs to the existing mobile reservoir within bacterial communities. However, reconstructing mARGs in metagenomic data from diverse ecosystems poses computational challenges, including genome fragment reconstruction (assembly), high-throughput annotation of MGEs, and identification of their association with ARGs. Recently, several bioinformatics tools have been developed to identify assembled fragments of plasmids, phages, and insertion sequence (IS) elements in metagenomic data. These methods can help in understanding the dissemination of mARGs. To streamline the process of identifying mARGs in multiple samples, we combined these tools in an automated high-throughput open-source pipeline, MetaMobilePicker, that identifies ARGs associated with plasmids, IS elements and phages, starting from short metagenomic sequencing reads. This pipeline was used to identify these three elements on a simplified simulated metagenome dataset, comprising whole genome sequences from seven clinically relevant bacterial species containing 55 ARGs, nine plasmids and five phages. The results demonstrated moderate precision for the identification of plasmids (0.57) and phages (0.71), and moderate sensitivity of identification of IS elements (0.58) and ARGs (0.70). In this study, we aim to assess the main causes of this moderate performance of the MGE prediction tools in a comprehensive manner. We conducted a systematic benchmark, considering metagenomic read coverage, contig length cutoffs and investigating the performance of the classification algorithms. Our analysis revealed that the metagenomic assembly process is the primary bottleneck when linking ARGs to identified MGEs in short-read metagenomics sequencing experiments rather than ARGs and MGEs identification by the different tools.

Introduction

Antimicrobial resistance (AMR) is an existing problem in veterinary and public health, with an increasing number of infections caused by resistant bacteria. Globally, in 2019, an estimated 1.27 million deaths were attributable to AMR (Murray et al., 2022). Antimicrobial resistance genes (ARGs) may result from mutations in bacterial genomes or alternatively be acquired from extrinsic sources. Acquired ARGs are frequently found on mobile genetic elements (MGEs) such as plasmids or transposons (Partridge et al., 2018). These elements can facilitate efficient spread by translocation via intragenic exchange between plasmids and chromosomes (Borowiak et al., 2017), or disseminate between genomes in inter- and intra-species horizontal gene transfer (HGT) events (Redondo-Salvo et al., 2020). AMR in terms of public health is tightly connected to the health of domesticated pets and livestock, as well as the environments humans share with these animals. These shared environments represent key crossover points by which resistant bacteria and ARGs can spread across various reservoirs, including clinical, agricultural and environmental reservoirs (Stanton et al., 2020). Consequently, AMR presents itself not only as a public health challenge, but also as a One Health issue (McEwen & Collignon, 2018; Collignon & McEwen, 2019; Despotovic et al., 2023).

Until now, studies on the spread of MGEs containing ARGs (mARGs) between different reservoirs like humans and animals have mostly been limited to clinically-relevant pathogenic bacteria involving single isolate cultures in combination with whole genome sequencing of single indicator organism isolates. Examples include studies on the dissemination of extended-spectrum beta-lactamase (ESBL)-producing Escherichia coli (Mughini-Gras et al., 2019), methicillin-resistant Staphylococcus aureus (MRSA) (Sieber et al., 2018) and colistin-resistant (i.e, mcr-1) E. coli (Wang et al., 2018). However, mARGs are not limited to pathogenic species; they are also present in commensal species (Despotovic et al., 2023) and can be transferred by several well-defined mechanisms such as conjugation or transduction (Hall, Brockhurst & Harrison, 2017). Conversely, genomes of non-pathogenic species in microbial communities are an important reservoir of mARGs that can spread to clinically-important species (Lee et al., 2020), where they can persist even in the absence of antimicrobial selection (Carroll & Wong, 2018). Therefore, ARGs spreading within and between complex ecosystems contribute to the eco-evolutionary dynamics of a microbial community, i.e., a microbiome (Coyte et al., 2022). It is therefore important to investigate the entire AMR reservoir, known as the resistome, that is associated with the detected mobilome within metagenomes of given microbiomes. The application of metagenomic shotgun sequencing of bacterial communities can help achieve this objective.

Computationally, the analysis of metagenomic data poses additional challenges when compared to the analysis of single bacterial isolate whole genome sequencing data. Metagenomic data typically consists of a larger number of sequencing reads, with lower per-genome coverage, while at the same time representing a higher taxonomic complexity. As a result, metagenomic analysis tools require more computational resources compared to whole genome sequencing tools. The difference in computational resources becomes apparent in the metagenomic assembly step. Benchmarks of different metagenomic assembly tools show that metagenomic assemblers vary in the use of computational resources. However, in general, well-performing metagenomic assemblers using variable k-mer sizes such as MetaSPAdes (Nurk et al., 2017), MEGAHIT (Li et al., 2015a) and IDBA-UD (Peng et al., 2012) tend to require the most resources, especially memory and runtime (Mendes et al., 2023; Zhang et al., 2023; Quainoo et al., 2017). In addition, the de Bruijn graph constructed by metagenomic assemblers has a higher complexity and is therefore more challenging to traverse than those obtained from single isolate whole genome sequencing assemblies. Similarly, the increase in the complexity and amount of data produced by metagenomic assembly can lead to an increase in resource usage for post-assembly processes such as functional annotation. Aside from computational resources, annotation of metagenomic assemblies is complicated by the scarcity of reference databases for uncultivated species and computational models (Liu et al., 2022). The data in reference databases for functional or taxonomic annotation is biased towards clinically relevant and culturable species and strains, whereas the majority of diversity in a metagenomic community, especially in anaerobic conditions such as the gut, consists of unculturable species (Bernard et al., 2018). Computational models built for the annotation of specific structures, like insertion sequences (IS) or plasmids, are trained on data which pose this same challenge, i.e., biases for clinically-relevant and culturable species. This can introduce a bias towards familiar data, akin to reference-based methods, even though no reference database is used. Lastly, due to MGEs containing many repetitive regions (Treangen & Salzberg, 2011; Oliveira et al., 2010; Che et al., 2021), assembly algorithms can have difficulty discerning different copies of repeat-rich parts of MGEs, resulting in collapsed contigs (Tørresen et al., 2019). Although mock communities are available, as well as tools to simulate datasets from genomic sequences, these options also can be biased towards clinically and culturable species, similar to reference databases. Hence, accurately estimating authentic taxonomic diversity and complexity in complex ecosystems poses a challenge with the existing tools and databases. Yet, in silico simulation of metagenomic data is important to establish a ground truth to benchmark analysis tools.

To investigate the spread of mARGs in metagenomic data, our study aimed to evaluate the effectiveness of several established tools in identifying key mobile genetic elements: plasmids, insertion sequences, and phages. We integrated these tools into a comprehensive, reproducible, modular and scalable open-source pipeline called MetaMobilePicker (http://metamobilepicker.nl). To validate the ability of these tools to identify mARGs in metagenomes, we simulated a simplified microbial community using genomes and plasmids from seven clinically relevant species. Our assessment revealed that the classification algorithms and annotation tools were not performing optimally in recovering mARGs effectively. In this study, we explored the reasons behind this suboptimal performance, and showed that the accuracy of detecting mobile genetic elements and ARGs was highly dependent on the biases introduced during metagenomic assembly.

Methods

MetaMobilePicker

MetaMobilePicker, available at http://metamobilepicker.nl, is an open-source software pipeline linking detected ARGs to MGEs in metagenomes. MetaMobilePicker was written in Python 3.10 using the workflow management system Snakemake (Mölder et al., 2021) and was designed to run primarily on high-performance computation infrastructures, as MetaSPAdes (Nurk et al., 2017) is resource-intensive (Zhang et al., 2023). Versions of required software are listed in Table S1. The pipeline is installable as a Python package or by using conda from the bioconda channel. The pipeline can be divided into four parts: preprocessing metagenomic short-read sequences, genome fragment reconstruction, ARG annotation, MGE identification and output construction linking ARGs to MGEs. An overview of the overall workflow is displayed in Fig. 1. In the preprocessing steps, sequencing reads are deduplicated for PCR artifacts and filtered for read quality and (host) contamination. This is done using the QC module of Metagenome-Atlas (Kieser et al., 2020). Next, a metagenome assembly is performed using metaSPAdes (Nurk et al., 2017) on high-quality reads with default parameters as specified in the configuration file. Very short contigs of length less than 1 kbp are discarded for their limited use in downstream analyses. MGEs are identified using PlasClass (Pellow, Mizrahi & Shamir, 2020), ISEScan (Xie & Tang, 2017) and DeepVirFinder (Ren et al., 2020), respectively. ARGs are then annotated using ABRicate (Seemann, 2023) on the ResFinder (Zankari et al., 2012) database. Finally, all output files are combined, linking ARGs to MGEs if present on the same contig.

Figure 1 Workflow of the MetaMobilePicker pipeline.

Colors indicate the different steps/modules in the pipeline. Light blue: preprocessing and assembly. Blue: MGE identification. Dark blue: ARG annotation. Orange: output construction. Software tools used are indicated in brackets. AMR: Antimicrobial resistance. MGE: Mobile Genetic Element. IS: Insertion Sequence. The pipeline is available at http://metamobilepicker.nl. For software references and versions see Table S1.

Validation using simulated data

In order to validate the results of the different tools included in MetaMobilePicker (PlasClass, ISEScan, DeepVirFinder and ABRicate), we constructed a simplified simulated metagenomic dataset. We selected the genomes of seven highly resistant bacterial species from the PATRIC database (Wattam et al., 2014 accessed on March 3rd 2021). For this dataset, we selected a representative completed genome from E. coli, S. aureus, Enterococcus faecalis, Mycobacterium tuberculosis, Salmonella enterica, Klebsiella pneumoniae and Acinetobacter baumannii. In addition to these genomes, five phage genomes specific to the selected genera were selected. Selected organisms and associated plasmids and phages can be found in Table 1. Using the selected genomes, we simulated a dataset of 20 million paired-end reads (10 M per end) of 150 bp using InSilicoSeq (Gourlé et al., 2019) using the MiSeq error profile. To test if 10 million simulated reads were sufficient to resolve the complexity of the dataset without introducing large gaps when comparing the metagenomic assembly to the reference genomes, we simulated 40 million reads per end and subsampled them at intervals of 5 million reads between 5 and 30 million. We used MetaQuast (Mikheenko, Saveliev & Gurevich, 2016) to calculate the genome completeness per genome. In order to simulate different relative abundances of different genomes, we generated abundance profiles for all species using a lognormal distribution. Additionally, a copy number for the plasmids was determined using a geometric distribution with probability P=min1,log10Lr where L is the length of the plasmid in bp, to simulate smaller plasmids having a higher probability of having a higher copy number than larger plasmids and r is a reduction factor based on the order of magnitude of the sequence length. For plasmids, we set r = 7 and for phages we set r = 5. This copy number was multiplied by the abundance of the corresponding genome, and the abundances were normalized to sum to one. The complete genome assemblies, plasmids and phages were annotated for ARGs using ABRicate with the ResFinder. Additionally, ISEScan was used to identify the present IS elements present in the sequences. To backtrace the contigs assembled during the pipeline’s run associated with the plasmids and phages, we used Minimap2 (Li, 2018) as part of the MetaQuast workflow. Contig alignments shorter than 65 bp and with identity lower than 95% were removed, as per MetaQuast default.

Table 1 Strain information for the selected genomes of the simulated dataset.

Genbank ID, sequence length in kilobasepairs and relative abundance per genomic unit. Relative abundance is established by determining a copy number based on a geometric distribution where smaller plasmids and phages have a higher probability of occurring more than once.

Species - strain - phage strain	Sequence type	Number of ARGs	Accession ID	Sequence size (kbp)	Relative abundance	
A. baumanii - BJAB0715 - LZ35	chromosome,
plasmid,
phage	1,
2,
0	NC_021733.1,
NC_021734.1,
NC_031117.1	4,002,
52,
45	0.02031,
0.02031,
0.04424	
E. faecalis - SCAID PHRX1-2018 - IME_EF3	chromosome,
plasmid,
phage	7,
4,
0	NZ_CP041877.1,
NZ_CP041878.1,
NC_023595.2	2,598,
98,
42	0.02882,
0.02882
0.02882	
E. coli - AR-0427	chromosome,
plasmid,
plasmid	1,
8,
0	NZ_CP044148.1,
NZ_CP044149.1,
NZ_CP044150.1	5,530,
75,
86	0.01734,
0.01734
0.01734	
K. pneumoniae - BJCFK909, NTUH-K2044-K1-1	chromosome,
plasmid,
plasmid,
plasmid,
plasmid,
phage	3,
5,
4,
0,
0,
0	CP034123.1,
CP034124.1,
CP034125.1,
CP034126.1,
CP034127.1,
NC_025418.1	5,471,
200,
110,
86,
6,
43	0.03900,
0.07800,
0.03900,
0.03900,
0.03900,
0.03900	
M. tuberculosis - Beijing-like/35049 - DaVinci	chromosome, phage	2,
0	NZ_CP017593.1,
JF937092.1	4,427,
52	0.01338,
0.01338	
S. aureus - CMRSA-6 - S24-1	chromosome,
phage	11,
0	NZ_CP027788.1,
NC_016565.1	3,044,
18	0.13598,
0.13598	
S. enterica - 74-1357	chromosome,
plasmid	1,
6	NZ_CP018642.1,
NZ_CP018643.1	4,698,
119	0.10244,
0.10244	

To assess the performance of plasmid and phage classification, we calculated precision, defined in Eq. (1), recall, defined in Eq. (2), and the F1-score, defined as the harmonic mean between precision and recall. To assess the performance of the annotation of IS elements and ARGs, we calculate the sensitivity as defined in Eq. (3). (1) Precision=CorrectlypredictedcontigsCorrectlypredictedcontigs+Falsepositives

(2) Recall=CorrectlypredictedcontigsCorrectlypredictedcontigs+Falsenegatives

(3) Sensitivity=CorrectlypredictedISelementsorARGsTotalnumberofISelementsorARGs.

Bases not covered in simulated data

To infer the number of bases not covered by the simulated reads per reference genome, we used BWA MEM (Li, 2013) to align the simulated reads to the reference genomes. Using Samtools (Li et al., 2009), we inferred the coverage depth at each position on the reference genomes, and considered bases with a depth of 0 as not covered, as no reads mapping to these bases were found in the dataset.

Results

Establishing a baseline to validate the performance of the detection of MGEs and mARGs in metagenomes

We used reference genomes of seven bacterial species, including nine plasmids and five genus-specific phages, to simulate metagenomic reads as a validation data set for the tools included in MetaMobilePicker (Table 1). To establish a baseline for the performances of the detection tools for IS elements, plasmids, phages and ARGs, we ran ISEScan, PlasClass, DeepVirFinder and ABRicate with the Resfinder database on the completely sequenced reference whole-genome sequences as available in RefSeq and cross-checked with the available NCBI annotation. This resulted in the detection of 417 IS elements in the reference genomes, distributed among the seven bacterial genomes with most IS elements detected in the E. coli chromosome (n = 78) and least in the smallest of the two E. coli plasmids (n = 3) (Fig. 2). PlasClass identified plasmids and chromosomes without errors and misclassified two phages as plasmids (out of a total of five) based on a cut-off score of 0.8. DeepVirfinder showed a perfect classification of phages and non-phages on the reference data. Finally, to establish a baseline for the AMR identification step on the finished reference genomes, we annotated the reference genomes using the ResFinder database with ABRicate. This identified a total of 55 predicted ARGs from 38 different families. Of these 55, 28 were only found once by ABRicate using the default parameters, five ARGs were found twice (tetB, ermA, ant(9)-Ia, aph(3′)-III and blaSHV-12), three ARGs were found three times (blaTEM, tetA and sul2), and two ARG (aph(6)-Id and aph(3″)-Ib) were found four times. Two genes (tet(B) and aph(3″)-Ib) were found with two different alleles. In the cases where multiple copies were identified, the ARGs were annotated in the same replicon in two cases, in different replicons of the same species in one case and to different replicons in two or more species in seven cases. Of these 55 ARGs, 29 genes are present on plasmids, belonging to 20 different gene families. ARGs present in the reference genomes, copy number and mobility of replicons are shown in Table S2.

Figure 2 Annotation plot of bacterial reference genomes in validation dataset.

Blue rectangles: insertion sequences. Orange arrows: antimicrobial resistance genes. (A) Klebsiella pneumoniae chromosome and three plasmids. (B) Acinetobacter baumannii chromosome and plasmid. (C) Enterococcus faecalis chromosome and plasmid. (D) Mycobacterium tuberculosis chromosome. (E) Salmonella enterica chromosome and plasmid. (F) Staphylococcus aureus chromosome. (G) Escherichia coli chromosome and plasmid.

Figure 3 Percentage of reads covering the reference genomes per sampled depth, averaged per species.

Solid lines indicate bacterial origin, dotted lines indicate phage origin.

Detection of MGEs and ARGs in the metagenome-assembled contigs has a moderate accuracy

From the 20 different genomic replicons from seven bacterial species, we simulated 20M metagenomic paired-end reads (10 million per end) using InSilicoSeq with different relative abundances (Table 1). The simulated dataset of 10M paired end reads was used to run MetaMobilePicker. After running the QC and assembly modules, the identification steps identify three types of MGEs: (i) plasmids, (ii) IS elements and (iii) phage sequences.

Metagenomes often have a read depth ranging from 10 million - 70 million per sample (Gweon et al., 2019), albeit with a far larger complexity than the validation data set simulated here. To test to what extent the read depth influenced the resolution of the genomes in our dataset, we simulated 40 million reads per end and subsampled them at 5, 10, 15, 20, 25 and 30 million reads and assembled them. We then used MetaQuast to calculate the completeness per genome. The average genome completion per sample depth is shown in Fig. 3. This figure shows that genome coverage completeness did not improve much after 10 million reads, with a maximum at 20 million reads, although the difference between 10 and 20 million reads was small (0.14%). We concluded that 10 million simulated reads were sufficient to resolve the complexity of this specific dataset. To maintain the defined relative abundances, and avoid this being skewed by the sub-sampling process, we used InSilicoSeq to simulate a definitive dataset with 10 million reads per end, totalling 20 million reads. Running the Metagenome Atlas (Kieser et al., 2020) QC module on these reads resulted in a total of 9,797,445 reads per end and 42,065 orphaned reads. The built-in MetaSPAdes assembly step resulted in 1,513 contigs, of which 887 contigs were larger than 1 kbp with an N50 of 82,245 and an L50 of 97. The total assembly length accumulated to 29,747,115 bp. Only contigs longer than 1 kbp length were used for the classification of MGEs and ARGs. To detect IS elements in the simulated metagenome validation data, we ran the ISEscan (Xie & Tang, 2017) module of MetaMobilePicker on the 887 contigs longer than 1 kbp. ISEScan uses a prebuilt HMM database to identify insertion sequences. In total, 241 IS elements in 144 metagenomic contigs were identified compared to 417 IS elements in the reference genomes, resulting in a sensitivity of 0.585. In order to predict plasmids in the metagenomic contigs, PlasClass as part of the MetaMobilePicker pipeline was applied to contigs larger than 1 kbp, with a cut-off classification score of 0.8. PlasClass uses a deep neural network and does not require a reference database during runtime. This resulted in 111 contigs predicted as plasmids (Fig. S1). Of these 111, 63 contigs were correctly predicted as plasmids and 756 contigs were correctly predicted as chromosomal. Additionally, 48 and 17 contigs were falsely predicted as plasmid and chromosomal, respectively, resulting in a precision of 0.57, recall of 0.79, and an F1 score of 0.66. Phages were predicted with the MetaMobilePicker module for DeepVirFinder (Ren et al., 2020), which uses a logistic regression model to predict phages and was tested as one of the best-performing phage prediction tools for metagenomics in a recent study (Ho et al., 2023). In order to measure the performance of DeepVirFinder on our metagenomic assembly, we cross-referenced the contigs predicted by DeepVirFinder with a classification score greater than 0.95, with the contigs that originated from the five phages in the dataset. This analysis showed 27 predictions, of which five originated from the phages added to our community. These five phage contigs were the full-length assemblies of these phages. Of the 22 contigs not originating from our added phages, functional annotation using blastx (Camacho et al., 2009) showed 14 having a direct link to phage DNA, most likely originating from prophages. Another four contigs were putatively linked to phages, containing hits not exclusively associated with phages. The remaining four contigs showed no clear link to phage DNA. The false positive contigs were notably shorter than the true positive phage contigs. Of the 22 false positives, three were larger than 10 kbp, with a median of 2,568 bp, indicating limited room for full-length genes and genomic context on many of the contigs. As we were unable to determine the number of prophage- or phage-related genes not identified by DeepVirFinder, we did not take the 18 phage-related genes into account when calculating the classification metrics. This resulted in a recall of 1.0, a precision of 0.555 and an F1-score of 0.713. To benchmark the performance of ARG annotation on our validation set, we cross-referenced the metagenomic hits with the genomic hits (Fig. 4). Of the 55 ARGs in the reference genomes, 38 were identified on the correct reference chromosome or plasmid in the metagenome-assembled contigs, resulting in a sensitivity of 0.697. For 16 of the remaining 17 genes, we identified the gene once, while they were present in multiple copies in the reference genomes. These 16 hits comprise 10 unique ARGs. The remaining hit was not found in the metagenomic contigs. Additionally, for the genes for which more than one allele was present (tet(B) and aph(3″)-Ib), only one allele was found. Only one sul2 gene was found more than once in the metagenomic assembly, which was found twice compared to an expected copy number of three in the reference genomes. The identified genes are displayed in Table S2. An overview of the classification metrics for plasmids and phages can be found in Table 2. An overview of the annotation metrics for IS and ARGs can be found in Table 3.

Figure 4 Circular graph displaying the mapping of antibiotic resistance genes (ARGs) (inner lines) on assembled contigs (light blue inner circle) and reference genomes (dark blue inner circle).

Second circle displays sequence origin and classification. Purple: (true positive) chromosomes. Green: (true positive) plasmids. Blue: (true positive) phages. Red: False positive plasmids. Yellow: false negative plasmids. Orange lines in the outer ring denote the location of the ARGs. Outer lines denote ARGs present more than once in the reference genomes.

Table 2 Precision, recall and F1-score for plasmid and phage classification.

Phage scores were corrected by discounting the false positive classifications containing genes directly related to phage DNA.

		Uncorrected			Corrected		
Benchmark	Precision	Recall	F1-score	Precision	Recall	F1-score	
Plasmids	0.59	0.79	0.66				
Phages	0.185	1	0.312	0.555	1	0.713	

Table 3 Sensitivity of annotation of insertion sequences (IS) and antimicrobial resistance genes (ARGs).

Sensitivity (ratio of true positive annotations to total number of IS and ARGs) was adjusted for ARGs and IS in the reference genomes that did not have a corresponding metagenomics assembled contig by removing those from the analysis.

	Sensitivity (total)	Sensitivity (assembled)	
IS	0.585	0.932	
ARGs	0.691	0.982	

To find potential causes for the observed percentage of IS elements that were not present in the contigs, for the low precision of the plasmid identification, and for the low number of annotated ARGs, we further investigated read coverage, the influence of contig length cutoffs and assembly of the different MGEs and ARGs.

Lacking read coverage explains a minority of missing MGEs in the assembled metagenome

To investigate whether missing or lower read coverage of the simulated data could explain the low precision observed for the identification of MGEs and mARGs in our metagenome validation data, we mapped the simulated reads against the reference genomes and validated the depth and breadth of coverage of the MGE annotated positions. This showed a varying range of percentage of bases not covered between 0.001% (E. coli plasmid) and 3.67% (E. faecalis plasmid), with an average of 0.48% missing bases with a standard deviation of 1.16% (Table S3). This indicates that only a small portion of reference bases was not covered by the simulated reads. None of these bases with missing coverage aligned to ARGs. The E. faecalis chromosome contains a total of 24,900 bases not covered, possibly caused by the low simulated abundance of this species. These are spread out over the genome, and cause eight IS elements to have a contiguous region of bases not covered of 100 bp. Additionally, two more IS elements have contiguous regions not covered larger than 50 bp (Table S4), both in the M. tuberculosis genome. The bases not covered predominantly align to the chromosome sequences, with only one phage (containing a singular base not covered), and two of the plasmids having bases not covered. These plasmids are the aforementioned E. faecalis plasmid, and one E. coli plasmid having a singular base not covered. In contrast, each of the seven chromosome sequences have bases not covered ranging from 178 (0.006% of the genome) to 24,900 (0.96% of the genome) bases. Lacking read coverage was therefore, at most, responsible for 3.67% of the overall recall for plasmids and for less than 0.01% of the overall recall for phages. Especially for IS in the E. faecalis chromosome, the missing read coverage caused difficulty in assembly and subsequent annotation. The IS elements containing regions not covered of 100 bp were not assembled into contigs in both the per-species assembly and the complete assembly.

Smaller contig length cutoffs do not improve recall of MGEs

We investigated if the contig length cutoff of 1 kbp, as applied by MetaMobilePicker could influence the recovery of MGEs. We analyzed the benchmark data set with a contig, length cutoff of 500 bp instead of MetaMobilePicker’s default of 1 kbp, annotated them as described previously and compared all outcomes. This resulted in 954 contigs (>500 bp), instead of 887 (>1 kbp). Of the additional contigs ranging between 500 and 1,000 bp, none contained an ARG. Furthermore, the contigs with a length between 500 and 1,000 bp that were predicted as plasmids were predominantly of chromosomal origin (11 of 15 contigs), increasing the number of false positives more than the number of true positives and lowering the precision for plasmid prediction from 0.57 (>1 kbp) to 0.54 (>500 bp). Likewise, four contigs with a length between 0.5 and 1 kbp were predicted as phages but were of chromosomal origin. Of these sequences, two were homologs of phage proteins, and the other two had a potential association with phage DNA. Therefore, all four of these sequences were subsequently not counted in the calculation of the phage metrics. This suggested that lowering the sequence length threshold increased the number of false positive predictions, which was not compensated by the increase of true positive predictions.

Complexity of metagenome assembly is not causal for assembly performance

When split by species, the prediction of plasmids by PlasClass exhibited a false-positive bias, especially for contigs from E. coli (Fig. 5). Of the 48 false positive plasmid predictions, 30 originated from the E. coli genome. The false positive predictions of phages did not show this bias, and no false positive annotations were found for ARGs and IS elements. To investigate if this overrepresentation was caused by a highly fragmented genome assembly, we separated the simulated reads per genome and used each set of reads as input for MetaMobilePicker separately. The resulting set of classified plasmid contigs was compared to the reference genome sequences and the metagenomic contigs using metaQuast. The classification scores for each species are displayed in Supplemental Table S5. Most notably, the number of false positive plasmid predictions originating from the E. coli genome for the single-species-only assembly was as high as in the metagenomic assembly with 30 false positive predictions. Comparing the false positive-predicted contigs from the single-species-only assembly to the metagenomic assembly showed 29 identical pairs, and one pair where the contig originating from the metagenomic assembly was 274 bp shorter (total length = 1,622 bp) than the contig from the reference sequences. For the other species, the single-species-only assembly also showed largely the same outcome as the metagenomic assembly with regard to plasmid-predicted contigs, with the exception of S. enterica. In S. enterica, the assembly of the plasmid improved notably in the single-species-only assembly (metagenome: 16 mapping contigs, average length 6.3 kbp, single-species-only: four mapping contigs, average length 28.1 kbp). However, as the S. enterica plasmid represented only one false positive prediction in the metagenome assembly, the S. enterica F1 score for metagenome or single-species only differed because of the higher degree of fragmentation of the metagenomic assembly (F1meta = 0.96, F1singlespecies = 0.8). Comparison of the wrongly predicted sequence lengths shows similar results (bpmeta = 9.8 kbp, bpsinglespecies = 11.0 kbp). Based on this, we note that the increased complexity of metagenomic assembly does not substantially influence the recovery of MGEs in the benchmark dataset.

Figure 5 Number of false positive plasmid predictions per replicon.

Assembly collapses identical insertion sequences and ARGs

To test if the low sensitivity of the identification of IS elements and ARGs was caused by difficulty assembling these elements, we first tested if all IS elements in the reference genomes were represented in the metagenomic contigs (Fig. S2). We mapped the metagenomic contigs larger than 1 kbp to the reference genomes. This showed that for 324/417 IS elements (77.70%) annotated in the reference genomes, a corresponding metagenomic contig was mapped, whereas a matching metagenomic contig was missing (less than 20% of the IS elements covered) for 93/417 IS elements present in the reference genomes (22.30%). This includes the 10 IS elements not assembled due to missing read coverage. Of the 324 reference IS elements with a corresponding metagenomic contig, 90 (27.80%) corresponded to 16 metagenomic contigs which mapped to multiple IS elements in the reference genomes. These 16/417 metagenomic contigs are therefore likely to represent assembly collapses. Furthermore, 16 IS elements annotated in the reference genomes (4.90%) corresponded only partially (between 20% and 80% of the IS elements covered) to a metagenomic contig. The remaining 221 IS elements in the reference genome (69.70%) corresponded to a metagenomic contig uniquely mapped to that IS element. To contrast the detection sensitivity of ISEscan before and after metagenome assembly, we also calculated the maximum achievable amount of IS elements detected in the metagenome-assembled contigs. To this end, we discounted all IS elements that were lost or only partially assembled during metagenome assembly. When taking only the IS elements into account that were present in the metagenomic contigs, and subtracting the duplications of the ambiguously mapped contigs, the maximal number of IS contigs ISEscan could potentially identify was 248. ISEscan detected 231, thereby achieving a sensitivity of 93.15%. From this, we conclude that collapse of IS elements during the assembly process was a major contributor to the sub-optimal identification sensitivity of IS elements.

As mentioned above, 16 of the ARGs in the reference genomes were not identified with the correct copy number. Further investigation of the metagenomic contigs mapped to the reference genomes showed that 13 of these genes did not have a corresponding metagenomic contig. The remaining three genes mapped partially to a contig but lacked annotation. This suggested that similar to the IS, ARGs were collapsed during the metagenomic assembly step.

Discussion

In this study, we aimed to identify and annotate ARGs linked to MGEs or mobile ARGs (mARGs) in metagenomic datasets. Using MetaMobilePicker, we were able to identify a large part of the MGE present in our simulated dataset (shown by the high recall of the classification and annotation of the MGEs) with an average true positive percentage of 91.3%, but also included many false positives (shown by the lower precision). Additionally, many mARGs and IS elements were not identified in the correct copy numbers. Most of the missed mARGs were lost in the process of metagenomic assembly, and there was no corresponding contig to all mARG copies originally present in the reference genomes.

Several pipelines for MGE identification have been published recently, such as PathoFact (de Nies et al., 2021), MobileElementFinder (Johansson et al., 2021), MGEFinder (Durrant et al., 2020), VRProfile2 (Wang et al., 2022), MetaCompare (Oh et al., 2018) and hgtSeq (Carpanzano et al., 2022). However, many of the existing tools (MobileElementFinder, MGEFinder, VRprofile2 and hgtSeq) were developed for sequencing data from bacterial isolates, often without being applicable to metagenomics (MobileElementFinder, MGEFinder, hgtSeq). Other tools, such as VRprofile2, include a use-case for long-read metagenomic datasets. All tools with the exception of MGEFinder and hgtseq, which use read alignment to a reference genome, require assembled contigs and are therefore prone to the same biases as described in this study. MetaMobilePicker provides a streamlined and reproducible workflow allowing processing of multiple samples with the same parameters and in parallel. This facilitates the possibility to compare the presence and absence of specific mARGs between samples and can lead to new insight into the flow of genes between different ecologies in a One Health context.

To quantify the performance of the tools included in MetaMobilePicker to identify mARGs, we used a computationally simulated dataset based on reference genomes. The composition of this dataset was an oversimplification of the complexity of real-world metagenomic samples with relevance to One Health, such as the human gut, livestock, and dust, in which thousands of species can be identified in a single sample (Bindari et al., 2021). Moreover, the species we selected are clinically relevant and present in many data-driven studies and were therefore likely to have been used in the training of the classification algorithms used to identify plasmids and phages. However, for validation and benchmarking, high-quality reference genomes are necessary to ensure the algorithms are validated on their ability to differentiate between the different origins of the sequence, rather than their ability to discover novel plasmids. The in silico community contained only seven bacterial species. This inflates the relative abundance of all of the genomes to levels not realistic in a real-world sample. To partially counteract this, we simulated a shallowly sequenced community with 10 million reads per end. Therefore, the biases uncovered in this analysis are likely to be even more pronounced in real metagenomic sequencing. Most notably, the resolution of within-sequence repeats (like IS elements) and between-sequence repeats (like similar plasmids or bacterial strains) will be amplified when sequencing the real-world complexity of bacterial communities (Olson et al., 2017).

On our validation data, plasmids were identified with high recall, but precision was lacking, due to an overrepresentation of, mainly, E. coli chromosomal fragments in the plasmid-prediction class. This overrepresentation might be due to a combination of factors. The E. coli reference genome contains the highest number of IS. Since IS are difficult to assemble, the E. coli genome was likely to be more fragmented than the other genomes. This was in addition to the fact that performing metagenomic assembly on only the subset of reads simulated from the E. coli genome resulted in an almost identical set of contigs with the same MGE classification errors. The higher degree of fragmentation can lead to a theoretical maximum of possible contiguity in the assembly. This, in turn, can cause difficulty for the classification tool in distinguishing contigs that can be in both a plasmid or chromosomal context, with limited context from flanking regions. The identification of phage sequences showed many false positives and, consequently, a low precision. We showed that this was primarily the result of prophage regions in the selected reference genomes. Although DeepVirFinder was not developed with the goal of identifying prophages, the high degree of fragmentation of the assembly caused the algorithm to classify these short contigs as phages. Short contigs contain limited amounts of information for the algorithm to determine that these contigs originated from bacterial reference sequences rather than from phages. Even after correcting for these prophage sequences, the precision of this classification remains low. This was caused by the much higher number of bacterial contigs compared to phage contigs. All five phages were assembled into a single contig each, which leaves little room for error when calculating prediction scores for 882 bacterial contigs. Conversely, identification of IS elements was shown to be highly accurate when only taking into account IS elements in the metagenomic assembly. However, comparing the IS elements identified to the expected numbers from the reference genomes, we observed that a large portion of IS elements was not found back in the final IS element annotation. This appears to be an issue with the metagenomic assembly being unable to differentiate between separate copies of the same IS element, rather than the ability of ISEScan to identify IS elements. The repetitiveness of IS elements complicates assembly using short reads because it can be problematic to identify a suitable path through the de Bruijn graph that connects the correct flanking regions of the IS elements. This can result in the termination of the path at the IS element and the collapse of multiple IS elements into one contig (Treangen & Salzberg, 2011). As IS elements are often found surrounding ARGs as part of transposable elements (Che et al., 2021), this process also complicates the assembly and annotation of mARGs.

Metagenomic assembly is an essential step in many short-read metagenomic workflows. Many tools exist to construct contigs from short metagenomic reads and extensive comparisons have been conducted on these different algorithms (Sczyrba et al., 2017; Zhang et al., 2023; Mendes et al., 2023). Although a number of benchmarks show the challenges with metagenomic assembly (i.e., Mendes et al., 2023), many of these challenges have not been described in the context of reconstruction of MGEs. De Bruijn graphs constructed for metagenomic assembly are highly complex due to the vast genetic diversity present in metagenomic samples. This is only increased in the reconstruction of MGEs, most notably plasmids and IS elements. Many MGEs consist of highly repetitive regions larger than the average k-mer used in the construction of the de Bruijn graph (Partridge et al., 2018). This complicates not only the assembly of these regions but also the assembly of genes shared between multiple replicons or genomes, as it is more difficult to reconstruct the correct sequences. Commonly, the metagenomic assembly is an intermediate product that is not used to generate the main results. The difference in the expected abundance of these plasmids makes it difficult to group them together with their genome of origin. This is also shown in our experiment, as many IS and plasmid sequences were not reconstructed with the correct copy numbers. Binning these results would not assign these to the correct genomes, either omitting several genomes, being put in a separate bin, or remaining unbinned, similarly as shown by Maguire et al. (2020) for genomic islands and plasmids.

In recent years, long-read sequencing techniques have proven useful in the reconstruction of MGEs like plasmids and IS elements (Berbers et al., 2020). Additionally, long-read metagenomics is an increasingly viable technique to capture the diversity of bacterial communities. Taxonomic profiling as well as the reconstruction of Metagenomic Assembled Genomes (MAGs) using long read techniques showed good results (Albertsen, 2023; Gounot et al., 2022) that can compete with short-read sequencing techniques. However, plasmids remain difficult to assemble even using long read metagenomic techniques, and high strain variation or bacterial species in low abundance in samples still provides challenges (Albertsen, 2023; Bouras et al., 2023). Using a hybrid solution combining the coverage depth of short reads and the read length of long reads can be a solution feasible for plasmid assembly, like in whole genome sequencing experiments, as well as techniques like Hi-C (Lieberman-Aiden et al., 2009) that can help distinguish between similar plasmids in different hosts (Cuscó et al., 2022).

Conclusions

In this study, we have assessed the role of metagenomic assembly in the identification of mobile genetic elements (MGEs) and antimicrobial resistance genes (ARGs). We conclude that the largest bottleneck in correctly identifying MGEs in a simulated metagenomic sample is the quality of the metagenomic assembly and, to a lesser extent annotation tools and sequence coverage.

Supplemental Information

Supplemental Information 1 Circular graph showing mappings of plasmid predicted contigs (inner lines) in assembled metagenomics reads (light blue inner circle) to IS on the reference genomes (dark blue inner circle)

Orange lines indicate unique plasmid mappings, blue lines indicate sequences mapping ambiguously to multiple locations on the reference genomes. Second circle displays sequence origin and classification. Purple: (True positive) chromosomes. Green: (True positive) plasmids. Blue: (True positive) phages. Red: false positive plasmids. Yellow: False negative plasmids.

Click here for additional data file.

Supplemental Information 2 Circular graph displaying mapping of IS (inner lines) in assembled metagenomics reads (light blue inner circle) to IS on the reference genomes (dark blue inner circle)

Dark lines indicate unique IS mappings, light lines indicate IS mapping ambiguously to multiple reference IS. Second circle displays sequence origin and classification. Purple: (True positive) chromosomes. Green: (True positive) plasmids. Blue: (True positive) phages. Red: false positive plasmids. Yellow: False negative plasmids. Blue lines in the outer ring denote missing IS. IS: insertion sequence.

Click here for additional data file.

Supplemental Information 3 Software versions and references used in MetaMobilePicker

Click here for additional data file.

Supplemental Information 4 ARGs in the reference community, the mobility type of the reference replicons they are found on, and the copy numbers in the reference genome and the metagenomics assembly

Click here for additional data file.

Supplemental Information 5 Number and percentage of bases not covered by simulated reads per replicon

Click here for additional data file.

Supplemental Information 6 Insertion sequences containing contiguous regions of uncovered bases (¿50 bp)

Click here for additional data file.

Supplemental Information 7 Plasmid precision, recall and F1 score per species when MetaMobilePicker was ran on the reads of each species individually and in the metagenomics sample

Click here for additional data file.

We thank Dr. Victória Pascal for designing the MetaMobilePicker logo and the rest of the bioinformatics group at the Medical Microbiology department of the UMC Utrecht for testing and improving the tool.

Additional Information and Declarations

Competing Interests

Author Contributions

Data Availability

The authors declare there are no competing interests.

Jesse J. Kerkvliet conceived and designed the experiments, performed the experiments, analyzed the data, prepared figures and/or tables, authored or reviewed drafts of the article, and approved the final draft.

Alex Bossers analyzed the data, authored or reviewed drafts of the article, and approved the final draft.

Jannigje G. Kers analyzed the data, authored or reviewed drafts of the article, and approved the final draft.

Rodrigo Meneses analyzed the data, authored or reviewed drafts of the article, and approved the final draft.

Rob Willems conceived and designed the experiments, authored or reviewed drafts of the article, and approved the final draft.

Anita C. Schürch conceived and designed the experiments, analyzed the data, authored or reviewed drafts of the article, and approved the final draft.

The following information was supplied regarding data availability:

The repository for the pipeline is available at MetaMobilePicker: http://metamobilepicker.nl.

The fastq files belonging to the simulated dataset is available at Zenodo: Kerkvliet, J., Bossers, A., Kers, J., Meneses, R., Willems, R., & Schürch, A. (2023). Simulated reads [Data set]. Zenodo. https://doi.org/10.5281/zenodo.8251712

All other data and scripts are available at Zenodo: Kerkvliet, J., Bossers, A., Kers, J., Meneses, R., Willems, R., & Schurch, A. (2023). Validation repository. Zenodo. https://doi.org/10.5281/zenodo.8256086

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
