# Peer review of "Metagenomic assembly is the main bottleneck in the identification of mobile genetic elements"

_PeerJ, doi:10.7717/peerj.16695_

## Round 0.1 · original submission · Major Revisions

All three reviewers put forward suggestions for modification. Please revise the manuscript carefully and answer the reviewer's questions.

Reviewer 1 ·

Basic reporting

Manuscript 89705v1
“Metagenomic assembly is the main bottleneck in the identification of mobile genetic elements” by Kerkvliet et al is a report of a bioinformatic workflow to identify specific types of sequences (mobile genetic elements that act as sources of antibiotic resistance) in metagenomic sequence data and the assessment of the pipeline performance.

The pipeline appears to be an effective process that will prove useful to researchers of metagenomes.


Fluent, well-written manuscript with occasional idiosyncratic or awkward sentences. (e.g, line 40-41)

Experimental design

METHODS
A bit unclear what part of the methods refer to the MetaMobilePicker and what parts are referring to the simulated validation testing, in part due to the verb tense. (lines114-119) Presumably, this paragraph all refers to the pipeline not the validation testing.

STRENGTHS
The pipeline is a practical, turn-key solution for researchers that want to scan metagenomic sequences.
The cross-tabulation between mobile genomic elements and antibiotic resistance genes has strong clinical and epidemiological relevance.
The validation testing pulled up a reasonable number of known antibiotic resistance genes and identified 10 that were present more than once in the test set.
The authors thoughtfully consider both phage sequence detection and bacterial prophage sequence detection and the overlap between them.

Validity of the findings

WEAKNESSES
There are other software packages available for alternative workflows and these were not evaluated side-by-side (though many are designed for analyzing single genomes not metagenomes).
As the authors correctly note, their test set is vastly less complex than many metagenomes.
Can we use a different term other than “precision” for the fraction of correctly classified contigs? “Precision” strongly connotes replication, which is not at all what that ratio calculates. I think of the ratio more like Type II error measurement of false positives. If I am understanding their ‘precision’ calculation correctly, the way false positives are represented in Fig. 5 seems a lot more intuitive. I don't really understand the use of the term 'recall' either, but at least it doesn't invoke other mathematical meanings....

Additional comments

MINOR COMMENTS
In lines 162-3, please refer the reader back to table 1. Unlike above, this paragraph sounds like all the genomes of the tested species were scanned, not just the seven representatives listed in Table1.

ISEScan is reputed to be quite sensitive so I am surprised that in the dataset of whole-genome sequenced RefSeq genomes that it only pulled up 417 insertion sequences, skewed toward E.coli. In addition to the difficulties (such as higher degree of fragmentation the authors noted and assembly weaknesses), Xie and colleagues trained the software on E. coli so perhaps ISEScan is a bit biased with regards to E. coli?
ABRicate should be written with the first three letters capitalized. (though github often omits capitals)
I am ambivalent about naming pipelines (as opposed to naming software products). But, if I routinely adopted this workflow, I am sure the pipeline naming convention would become appealing to me!
Fig. 3 – Inner circle, inner circle, second circle? In Figs 4 and 6, there is a faint outer circle, is there one in Fig. 3? Are the unbroken inner circles each a reference genome? Can we label these?

I think FigS1 is overall more useful than the information in Fig. 3 & 6; perhaps the choice of supplemental can be flipped?
Table Supp2 – most are single representatives per line. Can we include species names in the table?

I appreciated the discussion of metagenomics assembly as it relates to mobile genetic elements detection.

I didn’t try installing and running, but the package and instructions seem clear and complete. https://bioconda.github.io/recipes/metamobilepicker/README.html

OVERALL ASSESSMENT
A worthwhile and useful workflow, thoroughly analyzed. Recommend acceptance with minor revisions.

Reviewer 2 ·

Basic reporting

The value of metagenomic assembly for locating mobile genetic elements (MGEs) and antibiotic resistance genes (ARGs) was evaluated in this work. It is nicely written and arranged, and the figures are pertinent, good quality, clear, and understandable. Tables are suitable. However, there are some grammatical errors throughout the manuscript that need editing.

Experimental design

This study is well-designed and organized, allowing it to detect a sizable fraction of the MGEs included in our simulated dataset with a high true positive percentage utilizing MetaMobilePicker.

Validity of the findings

The data is trustworthy and important. The conclusion is quite apparent, and the study offers the authors sample data from which to form conclusions.

Additional comments

This is an interesting and important work aimed at identifying and annotating ARGs linked to MGEs or mobile ARGs (mARGs) in metagenomic datasets. This paper is well-
written, and it presents evidence that might be used as a new tool to ensure that public health crises are prioritized in research and development.
 Title: The title properly explains the purpose and objective of the article.
 Abstract: The abstract offers an accurate summary of the paper, and the language used in the abstract is easy to read and understand. However, it lacks clear objectives for
the study, so please include it for more clarity.
 Please include a few relevant keywords.
 Introduction:
 Introduction: The authors provide sufficient context on the subject.
 Line 36, 40, 42,…: Abbreviations should be defined at first mention and used consistently thereafter.
 Line 93-98: In these lines, the authors present the tools they used to investigate the spread of mARGs, while this should be included in the methods and not in the
introduction.
 Line 98-99: In these lines, the authors presented results, I suggest to be deleted, because the results were presented in detail in the results section.
 Line 99-102: These lines are related to the discussion of the results, not the introduction.
 The introduction lacks the purpose of the study, so please include it.
 Methods:
 Line 127: Abbreviations should be defined at first mention and used consistently thereafter.
 Line 148, 149: Correct the verb (replace is with are).
 Results: The findings are presented clearly, and the results are reliable.
 Lines 220: The numbers 1-9 are written as words.
 The figures are clear and comprehensible.
 Tables: Clear and comprehensive. However, it is preferable to include the titles in the upper margin of the tables, in addition to explaining the abbreviations in the lower
margin of the tables.
 Discussion:
 Line 232-233: There is no need to include the aim of the study in the discussion section, but rather the focus should be on discussing the results and comparing them with
previous studies.
 Line 387 : Correct the verb (replace were with was).
 Line 410: Replace Maguire et al. with Maguire et al. (2020).
 Line 411: Delete Maguire et al. (2020).
 Conclusions are clear, the study offers sample data for the authors to draw.
 Line 424-425: There is no need to include the aim of the study in the conclusions section.
 Line 426: Abbreviations should be defined at first mention and used consistently thereafter.
 Grammar: Need some revisions along the manuscript.

Reviewer 3 ·

Basic reporting

no comment

Experimental design

no comment

Validity of the findings

no comment

Additional comments

I only have one minor comment. At line 152 – “To compare classification and annotation performances, recall and sensitivity are regarded as the false positive percentage.” I think of recall and sensitivity as the true positive fraction, i.e. the fraction of observed (predicted) species that correctly identified, as defined earlier in this paragraph. The false positive percentages are the fractions that are identified but are not truly among the distinct known species in the metagenome. The true positive species plus the false positive species sum to the total number of species identified or predicted, which is not the same as the total number of species known from the simplified simulated metagenomic dataset (i.e., the simulated dataset based from reference genomes). By definition, earlier in this paragraph, recall and sensitivity are the fraction of true positive species, and this last sentence creates confusion, as the false positive percentage would be the total predicted minus the the true positive fraction (recall or precision value). If precision and recall decrease with an increasing number of false positive species, and the numerator is the number of true positive species, then the denominator must be the total number of observed (predicted) positive species from the data set (i.e., not the true number of positive species from the reference genomes), and the number of false positive species in buried in the denominator of the fraction. Could you provide a reference for this paragraph that clearly defines precision, recall and sensitivity.

---

## Round 0.2 · accepted · Accept

All three reviewers agreed to accept, and I agreed with them.

Please ensure you address the remaining items mentioned by Reviewer 3 in the proof stage.

Reviewer 1 ·

Basic reporting

The introduction is clearer, though it has picked up a misspelling of ‘extrinsic’. (line 38)

I thank the authors for pointing me to Krawczyk (2018) for an early use of precision to represent classification model true positive percentage. The original coinage seems unnecessarily confusing, but I can accept the authors propagating the practice.

Experimental design

No criticisms

Validity of the findings

No criticisms

Additional comments

The authors adequately addressed all my other concerns.

Reviewer 2 ·

Basic reporting

The authors have followed all suggestions and made all pertinent changes

Experimental design

The authors have followed all necessary suggestions and made all pertinent changes

Validity of the findings

The authors have followed all necessary suggestions and made all pertinent changes

Additional comments

The authors have made all necessary changes and implemented all previous recommendations. In addition, addressed all my concerns.

Reviewer 3 ·

Basic reporting

no comment

Experimental design

no comment

Validity of the findings

no comment

Additional comments

Again, I only have minor comments:
Thank you for providing details on the equations defining precision, recall and sensitivity. I was hoping you would provide references supporting the use of these terms in the context of assembly/annotation performance. Reviewer 1 also asked for confirmation that this are established nomenclature terms for annotation performance. I agree with reviewer 1, who well stated the challenge regarding connoting other mathematical meanings. Precision and sensitivity are particularly challenging for me, as those terms (in addition to others not used here, e.g., accuracy, agreement and specificity) can reference diagnostic test performance metrics. You provided references in your response to reviewer 1 regarding this use of precision in the literature, but did not reference these in the manuscript. Perhaps, it is unnecessary to add references to the manuscript to demonstrate the accepted application of these terms in the scope of these papers. I leave it to the authors to decide if providing an additional sentence with references to substantiate the use of these terms (precision, recall, sensitivity) in this context is necessary to help the novice to intermediate reader. (I personally needed to do a literature review search upon reading the first draft to think through your use of these terms). I do not insist on any addition beyond the added equations in the current version.

Grammar
Line 306 – comma after “was mapped”